# Comparative Analysis of Single-Cell RNA Sequencing Methods with and without Sample Multiplexing

**DOI:** 10.3390/ijms25073828

**Published:** 2024-03-29

**Authors:** Yi Xie, Huimei Chen, Vasuki Ranjani Chellamuthu, Ahmad bin Mohamed Lajam, Salvatore Albani, Andrea Hsiu Ling Low, Enrico Petretto, Jacques Behmoaras

**Affiliations:** 1Programme in Cardiovascular and Metabolic Disorders and Centre for Computational Biology, Duke-NUS Medical School, 8 College Road, Singapore 169857, Singapore; yi.xie@u.duke.nus.edu (Y.X.);; 2Translational Immunology Institute, SingHealth/Duke-NUS Academic Medical Centre, Academia, Singapore 169856, Singapore; vasuki.ranjani@visitor.nus.edu.sg (V.R.C.);; 3Department of Rheumatology and Immunology, Singapore General Hospital, Academia, Singapore 169856, Singapore; andrea.low.h.l@singhealth.com.sg; 4SingHealth Duke-NUS Medicine Academic Clinical Programme, Duke-NUS Medical School, Singapore 169857, Singapore; 5Institute for Big Data and Artificial Intelligence in Medicine, School of Science, China Pharmaceutical University, Nanjing 210009, China; 6Department of Immunology and Inflammation, Centre for Inflammatory Disease, Imperial College London, Hammersmith Hospital, London W12 0NN, UK

**Keywords:** single-cell RNA sequencing, multiplexing, SPLiT-seq, 10x, PBMC

## Abstract

Single-cell RNA sequencing (scRNA-seq) has emerged as a powerful technique for investigating biological heterogeneity at the single-cell level in human systems and model organisms. Recent advances in scRNA-seq have enabled the pooling of cells from multiple samples into single libraries, thereby increasing sample throughput while reducing technical batch effects, library preparation time, and the overall cost. However, a comparative analysis of scRNA-seq methods with and without sample multiplexing is lacking. In this study, we benchmarked methods from two representative platforms: Parse Biosciences (Parse; with sample multiplexing) and 10x Genomics (10x; without sample multiplexing). By using peripheral blood mononuclear cells (PBMCs) obtained from two healthy individuals, we demonstrate that demultiplexed scRNA-seq data obtained from Parse showed similar cell type frequencies compared to 10x data where samples were not multiplexed. Despite relatively lower cell capture affecting library preparation, Parse can detect rare cell types (e.g., plasmablasts and dendritic cells) which is likely due to its relatively higher sensitivity in gene detection. Moreover, a comparative analysis of transcript quantification between the two platforms revealed platform-specific distributions of gene length and GC content. These results offer guidance for researchers in designing high-throughput scRNA-seq studies.

## 1. Introduction

Single-cell RNA sequencing (scRNA-seq) has revolutionized the field of biomedical research by enabling the comprehensive profiling of mRNA expression levels at the single-cell level. This technology provides a means to unravel the inherent heterogeneity among cells, either through measuring proportional changes or alterations in gene expression [1]. Furthermore, recent advances in computational tools have facilitated the extraction of valuable insights from scRNA-seq data, including the exploration of cell differentiation trajectories and cell–cell communication events [2,3]. The exponential growth of scRNA-seq studies over the past decade has consolidated its wide usage as a powerful tool for addressing crucial questions in biomedical research. Nevertheless, the current limitations in sample-level throughput pose challenges for studies necessitating multiple conditions or samples, such as longitudinal studies. Traditional scRNA-seq protocols typically involve the preparation of a single sample in each library, which can inadvertently introduce batch effects when multiple samples are processed on different days and/or by different individuals. Errors arising from batch effects and the absence of biological replicates have been identified as significant factors contributing to false discoveries in scRNA-seq studies [4,5]. Therefore, there is a need for methods that can increase sample throughput to ensure appropriately powered investigations [6].

One promising approach to address these challenges is the pooling of labeled samples into a mixture and parallelly profiling all cells with the aim of reducing technical batch effects, library preparation time, and overall cost. Various sample multiplexing techniques have been deployed to scRNA-seq protocols with different levels of sample throughput [7]. One of these, Parse Bioscience v2, commercialized the split-pool ligation-based transcriptome sequencing (SPLiT-seq) technology [8], enabling parallel profiling of 1-96 samples with up to 1 million cells in a single experiment. SPLiT-seq uses four rounds of combinatorial barcoding to index fixed and permeabilized cells in parallel, without physically partitioning cells. The first round of barcoding serves the purpose of adding sample-level barcodes to cells. More specifically, each sample is distributed to a single well of a 96-well plate. Well-specific barcodes are appended to transcripts through an in-cell reverse transcription (RT) reaction. After this step, all the cells are pooled and redistributed to a new plate for the second round of barcoding. This split-pool-ligation procedure is repeated three times, resulting in a unique combination of three barcodes utilized as individual cell barcodes. Unique molecular identifiers (UMIs) are then added to each complementary DNA (cDNA) molecule in the third round to avoid amplification bias. After three rounds of barcoding, the cells are split into eight sub-libraries where library-specific barcodes are added and cDNA molecules are amplified. While commercially available plates from Parse Bioscience currently allow a maximum of 96 samples in a single plate, the implementation of the first round of barcoding in a 384-well plate can easily scale up the system to accommodate 384 samples [8]. Given the high-throughput potential of this platform, it is important to compare its performance with conventional scRNA-seq protocols, where samples are not multiplexed and transcriptomic profiles are generated independently for each sample.

Here, we systematically benchmarked Parse Bioscience v2 (hereinafter referred to as “Parse”) against the conventional scRNA-seq protocol 3′ v3.1 (hereinafter referred to as “10x”) from 10x Genomics Chromium by comparing the transcriptomic profiles of peripheral blood mononuclear cells (PBMCs). The 10x platform utilizes a droplet-based scRNA-seq protocol, where each individual cell is captured and enclosed with a barcoded bead in a droplet using a microfluidic system [9]. After reverse transcription occurring within each droplet, the droplets are merged by demulsification and the cDNA amplification is completed. We selected the 10x protocol for comparison because it is widely employed in scRNA-seq studies [10], and it has been extensively used to query human PBMCs for various research questions such as new cell states common to healthy individuals [11] or to specific to a certain disease [12] or even to aging-related immune hallmark signatures [13]. For this platform comparison, we applied both Parse and 10x to the same PBMC samples obtained from two ethnicity-matched healthy donors. The Parse replicates were multiplexed with nine other human PBMC samples, while the samples processed with the 10x protocol were not multiplexed, resulting in two independent libraries. We compared the transcriptomic profiles demultiplexed from Parse data with those obtained from a single 10x library. This allowed us to assess the ‘library efficiency’ (i.e., fraction of reads with valid barcodes depending on cell recovery), gene detection sensitivity, accuracy in gene expression quantification, and ability to recover biologically distinct cell clusters between the two platforms.

## 2. Results

### 2.1. Study Design and Metrics

We benchmarked protocols (Parse Evercode WT v2 and 10x 3′ v3.1) from the Parse and 10x platforms using PBMCs from two healthy donors (Figure 1A, Appendix A). Human PBMCs serve as an ideal model system for benchmarking due to their heterogeneity in cell sizes and total RNA content, which are key factors that affect the performance of scRNA-seq methods [14]. Furthermore, human PBMCs are a well-studied heterogenous system where the major immune cell types and their corresponding marker gene expression have been previously defined [11,15,16]. Human PBMCs from two healthy donors (designated here as H1 and H2) were prepared in a single batch and distributed into two aliquots. Aliquot 1 was used for 10x library preparation without multiplexing the H1 and H2 samples. Aliquot 2 was used for Parse library preparation by multiplexing H1 and H2 with nine other samples into a single library prep. All libraries were sequenced together to minimize differences in sequencing depth. For each platform, we aimed to collect ~10,000 cells per sample.

In order to perform a comparative assessment of the two platforms, we used four key metrics: library efficiency, gene detection sensitivity, accuracy in gene expression quantification, and ability to recover distinct cell clusters. Library efficiency is directly related to the number of input cells and sequencing depth used in scRNA-seq studies [14]. As such, the two key factors that can affect library efficiency are cell recovery rate and fraction of reads with valid barcodes. Platforms that achieve a higher cell recovery rate are advantageous for studies that rely upon valuable (i.e., difficult to obtain) samples. A lower fraction of valid reads usually indicates high background noise [17] and requires deeper sequencing (and relatively increased cost) to ensure optimal transcriptome coverage. Sensitivity and accuracy in transcript detection and quantification will influence the ability to assess cellular heterogeneity. Higher sensitivity in gene detection will influence the characterization of discrete cell clusters, potential cell states within clusters (i.e. sub-clusters) and cell–cell communications.

### 2.2. Comparison of Library Efficiency

One factor affecting the scRNA-seq efficiency is the fraction of reads with valid cell barcodes. Too many invalid reads indicate the presence of high background noise in the library [14]. The fraction of valid reads was ~85% for Parse and ~98% for 10x (Figure 1B). Valid reads were then mapped to the human genome to examine the distribution of reads throughout the genome. We observed that Parse had a higher proportion of intronic reads and lower proportion of exonic reads compared to 10x (Figure 1B). One possible explanation for this result is that the oligo-dT primers used in the 10x platform induced a biased priming towards exonic regions next to 3′ poly-A tails [18]. Parse reduced this bias by utilizing a mixture of oligo-dT and random hexamer primers [19]. The duplicate rate is calculated as the fraction of mapped, valid cell-barcoded, valid-UMI-attached reads that are not unique. We observed that the duplicate rate in 10x was 56.0% for sample H1 and 50.1% for H2, which were higher than in Parse with 38.2% for H1 and 34.9% for H2 (Appendix A). This is consistent with the lower read efficiency in Parse since more sequencing capacity is occupied by invalid reads, resulting in less duplicated reads being sequenced from the library.

Another significant factor in scRNA-seq efficiency metrics is the fraction of cells recovered in the data relative to the input, namely the cell capture efficiency. This is particularly important when one considers preparing libraries from biospecimens with a low number of cells. The number of valid cells recovered in each sample was directly obtained from the 10x and Parse preprocessing pipelines. Both methods use barcode ranking plots to identify barcodes with valid cells and barcodes derived from ambient RNA. We observed that, on average, ~53% of cells were recovered in 10x, and ~27% of cells were recovered in the Parse data (Figure 1C, Appendix A).

### 2.3. Sensitivity in Gene Detection

To compare the sensitivity of mRNA capture between the two methods, we down-sampled the sequencing reads to 20,000 per cell for each sample from 10x and Parse. Overall, we observed an ~1.2-fold increase in the number of detected genes in Parse, and median values of 1886 and 1984 genes in the 10x H1 and H2 samples, respectively, compared to 2319 and 2283 genes in the Parse H1 and H2 samples, respectively (Figure 2A, Appendix A). Note that within each platform, the two biological replicates showed a similar number of detected genes/UMIs, indicating high technical reproducibility for both 10x and Parse (Figure 2A, Appendix A). To better characterize the library complexity of the two methods, we quantified the number of detected genes/UMIs at varying sequencing depths. We observed that both platforms had a similar number of detected UMIs while Parse consistently detected more genes at varying sequencing depths (Figure 2B).

After removing low-quality cells (cells with low UMI/gene counts or high percentage of mitochondrial mRNA) and doublets, we recovered 6920 and 7020 cells for samples H1 and H2, respectively, in the 10x data. Similarly, the Parse dataset yielded 9819 and 8259 cells for samples H1 and H2, respectively (Appendix A). Cell type identities were obtained by consensus annotation using cell-type markers and reference datasets (see Section 4; Figure 2C,D; Appendix A). The sensitivity of mRNA capture was determined separately for the major cell types in human PBMCs, namely T cells, Natural Killer (NK) cells, monocytes, and B cells. Across all cell types, Parse consistently detected a higher number of transcripts, with the exception of monocytes where the trend was reversed (Figure 2E). Specifically, within the monocyte population, the 10x dataset detected an additional 187 genes compared to Parse (Appendix A). This discrepancy may be due to the improved performance of the 10x protocol when applied to cells with relatively high total RNA quantities [20].

A dropout event is the phenomenon whereby genes with low expression levels quantified in one cell are not detected in another cell, leading to challenges in differential expression analysis [21]. Here, we estimated the ability of each platform to detect genes at various expression levels by calculating dropout probabilities. We observed that 10x and Parse had similar abilities to detect genes at varying expression levels (Figure 2F).

### 2.4. Bias in Gene Quantification

Genes expressed in less than eight cells across all samples were considered to be lowly expressed and removed from our dataset. In sample H1, there were 25,822 commonly expressed genes between the two platforms, 2035 Parse-specific genes, and 1702 10x-specific genes (Appendix A). Similarly, for sample H2, we found 1814 genes uniquely detected in Parse, 1739 genes uniquely detected in 10x, and 26,006 genes detected by both platforms (Appendix A). We investigated the abundance of various transcript subtypes in the uniquely detected genes from each method, and found that processed pseudogenes were specifically enriched in the 10x data while unprocessed pseudogenes were enriched in the Parse data (Appendix A). This can be attributed to the fact that unprocessed pseudogenes have introns that are preferentially targeted by the random hexamers used in the Parse protocol. Conversely, processed pseudogenes, characterized by their poly-A tails, are more likely to be detected by the oligo-dT primers used during the 10x method.

We then assessed the similarity of the gene expression profiles detected by the two methods using a correlation analysis. We randomly kept 7000 cells for each sample in our down-sampled dataset (20,000 reads/cell). A total of 24,599 genes that were commonly expressed in all samples were used for the correlation analysis. For each technical replicate, the Pearson’s correlation coefficient (r) was calculated between the two platforms based on the average UMI counts across all cells (Figure 3A). We observed moderate variations between the two platforms with a Pearson’s r of 0.62 for H1 and 0.66 for H2. These values are similar to previously reported comparisons between 10x and other platforms where Pearson’s r ranged from 0.5 to 0.7 [20]. Based on these findings, we concluded that the expression of genes is largely conserved across the two platforms.

To characterize variance between the transcriptomes depicted by two platforms, we performed dimensionality reduction with principal component analysis (PCA) and Uniform Manifold Approximation and Projection (UMAP). We observed that cells formed distinct cell-type clusters including T cells, NK cells, B cells and monocytes. Within each cell-type cluster, the cells were further clustered by their platform of origin, indicating technical differences between the two platforms in gene quantification (Figure 3B). Potential biases in gene quantification could be related to three major factors: gene expression abundance, gene length and GC content [20,22,23]. We examined the influence of these factors by selecting marker genes for each method and compared the distribution of these factors between the two methods. We found that 10x had a higher sensitivity for shorter genes and genes with a higher GC content, which is consistent with previous findings [18], while Parse had a higher sensitivity for longer genes and genes with a lower GC content (Figure 3C, Appendix A).

Considering the distinct RNA enrichment strategies employed by Parse and 10x, we compared the distribution of transcripts across various RNA subtypes by analyzing the proportions of UMIs mapped to different RNA subtypes in each method. We observed that approximately 6–9% of all detected transcripts by both methods were from non-coding genes, with long non-coding RNAs accounting for 8.19% (H1) and 8.30% (H2) in Parse, which was higher than in 10x with 6.32% (H1) and 6.71% (H2) (Figure 3D, Appendix A). Protein-coding genes accounted for 90.88% (H1) and 90.20% (H2) of all detected transcripts for 10x and 88.65% (H1) and 88.59% (H2) for Parse (Figure 3D). Among the protein-coding genes, the proportion of UMIs mapped to transcription factor (TF) genes were 4.78% (H1) and 4.89% (H2) for 10x, which were lower than for Parse with 9.40% for H1 and 9.47% for H2, which is in line with Parse’s higher sensitivity in detecting lowly expressed genes, including those encoding TF that are typically lowly expressed (Figure 3D). Furthermore, our observations indicate that both the 10x and Parse platforms exhibited a comparable quantity of ribosomal protein-coding genes (RB genes), with 424 and 425 genes detected in the two samples in the 10x data, and 316 and 319 genes detected in the two samples in the Parse data. However, the expression levels of these genes differed significantly between the two platforms. Specifically, the proportion of UMIs mapped to ribosomal protein-coding genes varied from 0.4% to 66.0% in 10x, with a median value of 31.4%. In Parse, the range was dramatically less; from 0% to 5.3%, with a median value of 1.0% (Figure 3D). We also compared other scRNA-seq platforms and found a range of enrichment for ribosomal protein-coding genes in PBMCs (Appendix A).

### 2.5. Ability to Capture and Characterize Cell Type Diversity

Identifying distinct cell types within a heterogeneous population is a fundamental step in analyzing scRNA-seq data. It is essential for identifying rare cell types, transitional cell states, and functionally significant cell subpopulations. Additionally, it facilitates advanced analyses such as differential gene expression between experimental conditions and the investigation of cell–cell communication mechanisms. Cell type identification is typically accomplished using two main methods: manual annotation and reference-based scoring. Our cell type annotation was achieved by harmonizing results from these two methods (see Section 4). We observed that major immune cell types (T cells, B cells, NK cells, and monocytes) were identified by both platforms with consistent proportions (Figure 4A, Appendix A), which are also aligned with the cell type proportions previously reported in scRNA-seq studies using human PBMCs [24,25]. Rare cell types (abundance < 1%), such as dendritic cells (DCs), plasmablasts, hematopoietic stem and progenitor cells (HSPC), and proliferating cells, were identified by Parse only, which is consistent with the higher gene capture sensitivity resulting in improved cell cluster separation (Figure 4A).

To further illustrate the ability of these two platforms to chart cell type heterogeneity, we utilized different metrics to quantify the ability to distinguish and recover cell types with manual annotation or reference-based scoring methods. For the manual annotation, we used average silhouette width (ASW) scores to evaluate the clustering quality and compared the expression of marker genes. For the reference-based method, the area under the receiver operating characteristic curve (AUC) was applied to quantify the platforms’ relative power to distinguish and recover cell types. These metrics enabled us to quantitatively assess the effectiveness of the two platforms in capturing and characterizing cell type diversity.

During manual annotation, cells with similar transcriptomic profiles were grouped into clusters based on known marker genes that are specifically and highly expressed in particular cell types. The clustering quality of the two datasets was evaluated by calculating the average silhouette width (ASW) score, a commonly used measure for assessing data partitioning quality [26]. Consistent with its higher gene capture sensitivity, the Parse platform exhibited higher ASW scores at various resolutions in both samples (Figure 4B). This finding was supported by Parse’s ability to effectively identify rare cell types (Figure 4A). However, the expression levels of some cell-type-marker genes were significantly lower in the Parse dataset when compared to 10x (Figure 4C). When analyzed by Parse, marker genes like *CD3D*, *CD3G*, *CD3E*, and *NKG7* were found to be expressed in less than 50% of cells within their respective cell types. A gene expression correlation analysis between the two platforms indicated that these specific cell-type markers were detected in relatively lower abundances in Parse (Figure 4D, Appendix A), which compromises the Parse dataset’s ability to accurately annotate cell types using these markers. We have previously shown that Parse exhibited a gene quantification bias towards longer genes and genes with a low GC content (Figure 3C, Appendix A). Given that these cell-type-marker genes (*NKG7*, *S100A9*, *CD79A*, *CD3D*, *CD3E* and *CD3G*) have a relatively shorter length and higher GC content, we can infer that their lower detection may be due to Parse’s bias regarding gene length and GC content (Appendix A).

We further assessed the ability of the two platforms to recover cell types by reference-based scoring. Cell-type-specific gene signatures were derived from a reference dataset consisting of bulk RNA-seq samples generated from sorted and purified immune cell populations [27]. The area under the receiver operating characteristic curve (AUC) was calculated for each cluster to estimate how well the cells in that cluster can be distinguished from the remaining cells (See Section 4). The AUC summarized the performance of each cell-type-specific gene signature in separating a cluster of cells from the rest of the cells, with higher AUC values indicating better separability. We observed that Parse exhibited higher AUC values in T cells and B cells and lower AUC values in monocytes (Figure 4E). This observation is aligned with the trend of gene detection sensitivity in the two platforms, where a higher number of detected genes contributes to higher accuracy in identifying cell types.

## 3. Discussion

We conducted a comprehensive comparison between the widely used 10x platform and Parse Biosciences platform, a relatively newly developed scRNA-seq platform that utilizes sample multiplexing and parallel profiling of large number of cells. We considered two independent replicates for the Parse and 10x platforms, which were derived from the same healthy individuals, H1 and H2. For the Parse data, we multiplexed H1 and H2 with another nine human PBMC samples. By doing so, we maximized the capacity of the kit and allowed for a demultiplexed sample comparison with 10x. In general, both platforms produced high-quality data with the recovery of the desired number of cells, genes, and cell type clusters. Overall, the data quality was largely consistent across the two biological replicates within each platform. In our analysis, we compared the relative strengths and weaknesses of each platform and assessed the impact of key features considered for most scRNA-seq studies, including library efficiency, sensitivity in gene detection, potential biases in gene quantification, and the capability to identify cell types accurately.

We selected peripheral blood mononuclear cells for our platform benchmarking since they (i) produce good yields, (ii) are relatively easy to isolate, and (iii) the number of viable cells sequenced is generally high (>70%) after freezing. These factors minimize the potential bias introduced during sample collection and preprocessing, allowing us to focus on the platform comparison. Moreover, PBMCs serve as a good reference cell system for our comparative analysis since they have been extensively studied with regard to their cell type composition and gene expression patterns [11,15,16].

When compared with the 10x platform, Parse exhibited a two-fold lower cell recovery rate and ~13% fewer valid reads. This difference in library efficiency can be attributed to the distinct designs of the methodologies between the two platforms. In the 10x platform, a single droplet in the microfluidic channels aims to encapsulate one bead and a single cell. Cell lysis, barcoding, and reverse transcription all occur within this droplet. The number of cells in a droplet follows a super-Poissonian distribution, resulting in ~80% bead occupancy and a cell recovery rate of about 50% [16]. On the other hand, Parse does not physically partition cells; instead, it fixes and permeabilizes all cells in a single mixture. Barcoding and reverse transcription are performed through four rounds of cell splitting and pooling in a multi-well plate. While this iterative process enhances sample throughput, it may compromise the efficiency of library preparation. The repeated transfer of cells between wells and tubes increases the likelihood of cell loss during the procedure. The mechanical forces involved in this process can result in a higher rate of cell breakage, leading to the removal of these cells as background noise during the washing steps. These factors may explain why in Parse, approximately twice the number of cells needs to be loaded in order to obtain a comparable number of cells recovered by the 10x system. Despite the lower cell recovery rate in its libraries, Parse had fewer cells discarded in the quality control step, which can be attributed, for the greatest part, to a lower multiplet rate in the Parse platform compared with 10x. The 10x platform employs a single reverse transcription step to attach all barcodes, whereas Parse involves one reverse transcription step and three subsequent ligation steps. The inclusion of these four sequential reactions in Parse can increase the likelihood of barcodes failing to attach to the cDNA molecules. This disparity between single and sequential barcoding could explain why Parse exhibited a relatively higher fraction of reads without valid barcodes. Consequently, these invalid reads require an increased sequencing depth for Parse libraries in order to ensure optimal transcriptome coverage and accuracy of gene expression quantification.

Our analysis revealed a significantly higher proportion of ribosomal protein-coding genes in the 10x data when compared with Parse. Notably, these findings are not unique to our study, as similar observations have been reported when comparing SMART-seq and 10x platforms, whereby 10x data were enriched (2.6–7.2 fold) for ribosomal transcripts [28]. We expanded this comparative analysis to include a publicly available PBMC dataset generated using 12 distinct scRNA-seq protocols/platforms [20]. The proportion of ribosomal protein-coding genes was the highest in 10x, although other non-droplet-based platforms also showed a relatively high proportion (Appendix A). The reasons behind this platform-specific bias in ribosomal protein-coding gene enrichment require further investigations. By comparing the distribution of the GC content of marker genes in each platform against all genes, no distinct patterns were observed for IFC-based, droplet-based, plate-based, or nanowell-based methods (Appendix A). We observed that Drop-seq was better at detecting genes with a lower GC content, which is in line with previous reports [18]. Notably, the 10x platform with version 2 chemistry demonstrated a bias towards lower-GC-content genes, contrary to what we observed in our libraries constructed with version 3 chemistry (Appendix A). This may indicate a chemistry-influenced gene quantification bias in the 10x platform.

Gene detection rates can be increased by achieving greater sequencing depths. To compare the gene detection sensitivity between the two platforms in an unbiased way, the sequencing depth was standardized to 20,000 reads per cell for each sample. Our observations revealed that Parse detected approximately 1.2-fold more genes than 10x in T cells, NK cells, and B cells, but not in monocytes. This difference can be attributed to the fact that the sensitivity of the 10x platform relies on mRNA quantities present in cells [20]. The relatively low mRNA levels of T cells, NK cells, and B cells [29] affect the performance of 10x in detecting lowly expressed genes. Conversely, monocytes provide relatively higher RNA quantities [29] for 10x, thereby enabling enhanced gene detection performance. Generally, in order to have a higher sensitivity for detecting lowly expressed genes, reverse transcription chemistries can be improved for capturing low abundance mRNAs.

A higher gene detection sensitivity can greatly benefit downstream analyses of scRNA-seq data [14]. With Parse, we observed higher clustering performance and greater power in distinguishing cell types with cell type-specific gene signatures, and in detecting rare cell populations. For 10x, improving reverse transcription sensitivity as mentioned in last paragraph, or enhancing the depth of sequencing to improve library complexity may improve its ability to detect rare cell populations. Various computational methods [30,31], such as incorporating pathway information in deep learning models for better clustering or integrating with other scRNA-seq or spatial transcriptomics data, will also help in rare cell population identification. However, known cell-type-marker genes were lowly expressed in the Parse data compared to the 10x data. This may be attributed to Parse’s tendency to detect longer and low-GC-content genes, as opposed to 10x. Therefore, when annotating cell types in Parse data, an alternative approach such as mapping reference datasets with known cell type information [32] may be more suitable than relying solely on a few specific marker genes.

Both platforms have distinctive features that could be combined. For example, the microfluidic system used in 10x can be combined with the combinatorial barcoding techniques used in Parse to achieve high sample throughput while maintaining an optimal cell recovery rate. We previously highlighted that Parse, although being able to multiplex up to 96 samples, displayed relatively low cell recovery rates due to multiple rounds of cell splitting, pooling and washing. Cells with valid barcodes may get lost/damaged and subsequently removed as part of background RNAs during the washing step. We hypothesize that encapsulating cells with beads can reduce this potential loss since we observed that the 10x data showed a relatively higher cell recovery rate. Cell barcodes can be designed on beads in microfluidic systems to achieve similar combinatorial barcoding results as Parse.

Despite some differences between the two platforms, our analyses suggest that the scRNA-seq data demultiplexed from Parse have comparable quality to those obtained by a single 10x library. We provide a concise high-level summary of the relative strengths and weaknesses of each platform in Table 1. While it is not our aim to advise on the use of a specific platform, we believe these analyses and findings can be helpful in facilitating the design of scRNA-seq studies that require multiplexing.

## 4. Materials and Methods

### 4.1. Peripheral Blood Mononuclear Cell (PBMC) Collection

A standardized protocol was followed for isolating peripheral blood mononuclear cells (PBMCs) from venous blood samples (up to 30 mL) collected in ethylenediaminetetraacetic acid (EDTA)-treated vacutainer tubes after receiving the participant’s informed consent. Briefly, whole blood was centrifuged at 596× *g* for 15 min to isolate the plasma. Subsequently, the cellular fraction was diluted with phosphate-buffered saline (PBS) and layered onto Ficoll-Paque Plus (Cytiva (Uppsala, Sweden), #17144003) following standard density gradient centrifugation methods as per the manufacturer’s instructions. The collected PBMC layers were then subjected to red blood cell lysis using Ammonium-Chloride-Potassium (ACK) lysis buffer (Lonza Bioscience (Basel, Switzerland), #BP10-548E). The cell numbers were counted by staining with trypan blue. The isolated PBMCs were suspended in a freezing media composed of 90% fetal bovine serum (FBS, Gibco (New York, NY, USA), #16140-071) and 10% dimethyl sulfoxide (DMSO) and then cryopreserved in liquid N_2_ until use.

### 4.2. Single-Cell Suspension Preparation

The PBMCs were kept cryopreserved for 2 weeks and rapidly thawed in a 37 °C water bath until pea-sized frozen cryopreserved cells remained. Thawed PBMCs were resuspended by adding 12 mL of 37 °C pre-warmed 10% human serum media (Human Serum, Sigma Aldrich (Burlington, MA, USA), # H4522; RPMI 1640 Medium, Gibco, # 11875-093; Penicillin–Streptomycin–Glutamine, Gibco (NY, USA), # 10378-016) and were centrifuged at 500× *g* for 5 min at room temperature. The cell pellet was resuspended in 0.5 mL of pre-warmed human serum media and rested in a 37 °C, 5% CO_2_ incubator for 30 min. After resting, the cell suspension was passed through a 35 µm Falcon cell strainer (Falcon (Mexico City, Mexico), # 352235), and counted by the trypan blue staining method. Single live PBMCs were selected using fluorescence-activated cell sorting (FACS) with gating based on forward scatter (FSC), side scatter (SSC), and DAPI dye (Thermo Fisher (Waltham, MA, USA), #D1306). The purified PBMCs were then resuspended with 1 mL of 1X PBS containing 1% bovine serum albumin (BSA, Sigma Aldrich (MA, USA), # B2518).

For the 10x Genomics platform, the cell count for each sample was determined using the TC20™ Automated Cell Counter (Sigma Aldrich). Approximately 16,000 cells per sample were used for library construction, utilizing the Chromium Next GEM Automated Single Cell 3′ Reagent Kits (10x Genomics, Pleasanton, CA, USA). For the Parse Biosciences platform, the cells were prepared using the Fixation Kit (Parse Biosciences, Seattle, WA, USA) according to the manufacturer’s protocol. After fixation, the cell count was determined using the TC20™ Automated Cell Counter (Sigma Aldrich). A total of 390,000 cells from 11 donors was used, resulting in an average of approximately 35,500 cells per donor. These cells were subjected to the Single Cell Whole Transcriptome Kit v2 (Parse Biosciences, Seattle, WA, USA) for library construction. The scRNA-seq libraries from 10x Genomics and Parse Biosciences platforms were sequenced using the Illumina NovaSeq 6000 platform, generating 200 bp paired-end reads as follows: for 10x, 203,134,663 (H1) and 254,103, 388 (H2); for Parse, 297,237,012 (H1) and 262,086,443 (H2) (Appendix A).

### 4.3. Parse and 10x Data Downsampling

For the Parse data, fastq files from 8 sub-libraries were demultiplexed into 11 samples using the *split-pipe* pipeline (v1.0.4p) from Parse Biosciences (https://support.parsebiosciences.com/hc/en-us/categories/360004765711-Computational-Support, assessed on 24 November 2023). Demultiplexed fastq files from samples H1 and H2 were used for downsampling. For the 10x data, each sample was prepared and sequenced in a single library. We directly used these two fastq files for downsampling. Read downsampling was achieved for each sample by using *seqtk* (v1.3-r106) with command ‘*seqtk sample*’. The random seed was fixed at 100 with argument *-s*.

### 4.4. Parse and 10x Data Preprocessing and Quality Control

The same human reference genome, GRCh38/hg38, was used to map and quantify the gene expression in the 10x and Parse data. The human genome reference files were downloaded from the Ensembl database (https://ftp.ensembl.org/pub/release-108/fasta/homo_sapiens/dna/Homo_sapiens.GRCh38.dna.primary_assembly.fa.gz, assessed on 15 December 2023; https://ftp.ensembl.org/pub/release-108/gtf/homo_sapiens/Homo_sapiens.GRCh38.108.gtf.gz, assessed on 15 December 2023). For the 10x data, read mapping and gene expression quantification were performed using *CellRanger* (v7.1.0) [16] using the default parameters except for *--include-introns=true*. For the Parse data, *split-pipe* (v1.0.4p) was used with the following parameters: *--mode all*; *--chemistry v2*; *--kit WT*; *--kit_score_skip*; *--no_allwell*.

Gene expression matrices were loaded into the R package Seurat (4.3.0) [33] for quality control and downstream analyses. We removed cells fulfilling any of these criteria: (1) number of UMIs ≤ 1500, (2) number of UMIs ≥ 20,000, (3) number of genes ≤ 500, and (4) percentage of mitochondrial genes ≥ 15%. Cell doublets were detected and removed using the R package *DoubletFinder* (2.0.3) [34]. Genes expressed in less than 8 cells were discarded. Gene counts were then normalized and transformed with *NormalizeData()*, which normalizes the total counts for each cell by the total counts, multiplies this by a scale factor of 10,000, and log-transforms the result.

### 4.5. Cell Type Annotation

Cell type annotation was performed separately for the Parse and 10x datasets using the same procedure. First, highly variable genes (HVGs) were obtained using the *FindVariableFeatures()* function with the default parameters. The top 2000 HVGs, which were arranged in decreasing order based on dispersion, were selected as the input for downstream analyses. The data were then scaled using the *ScaleData()* function with parameter *vars.to.regress = c(“nFeature_RNA”, “percent.mt”)*. Dimension reduction was performed using the *RunPCA()* and *RunUMAP(dims=1:20)* functions. The top 20 principal components (PCs) were used to construct nearest-neighbor graphs and identify cell clusters using the *FindNeighbors()* and *FindClusters()* functions of the *Seurat* (4.3.0) R package.

Cell type identity was confirmed using (1) consensus annotation from manual annotations with markers obtained from [14,35,36] and (2) reference-based annotations with the *SingleR* (2.0.0) R package [37]. For manual annotation, 12 major cell types were identified: CD4+ T cells (*CD3D+CD4+*), CD8+ T cells (*CD3D+CD8A+*), MAIT cells (*SLC4A10+*), NK cells (*NKG7+NCAM1+*), CD14+ monocytes (*LYZ+CD14+*), CD16+ monocytes (*LYZ+FCGR3A+*), cDCs (*FCER1A+*), pDCs (*CLEC4C+*), B cells (*MS4A1+*), plasmablasts (*MZB1+*), proliferating cells (*MKI67+*), and HSPCs (*CD34+*) (Appendix A). For reference-based annotations, the *SingleR()* function was used to calculate the gene expression correlation with the reference dataset from Monaco et al. [29] using the default parameters.

A consensus cell-type annotation between the manual and reference-based annotations was obtained using following steps: (1) compare manual and *SingleR* annotations, and if they are identical, leave the cell type annotation as it is; (2) if one of the two annotations has a higher resolution (i.e., a cell type (e.g., CD4+ T cell) is a subset of another cell type (e.g., T cell)), use the annotation with the higher resolution; (3) if the two annotations are still inconsistent, use canonical markers to re-annotate the cell; and (4) remove cells that express conflicting cell-type markers (e.g., cells co-expressing CD3D+ T cell and MS4A1+ B cell markers) as potential doublets.

### 4.6. Library Efficiency

The number of reads without valid barcodes were obtained and summarized by *split-pipe* (v1.0.4p) and *CellRanger* (7.1.0) separately, and divided by total number of reads to obtain the fraction of valid reads for each sample. *QualiMap* (2.2.2) [38] was used to quantify reads uniquely mapped to exons, introns, intergenic, and overlapping-exon regions between any two overlapping genes. For each sample, the cell capture efficiency was calculated by dividing the number of cells output from *split-pipe* (v1.0.4p) and *CellRanger* (7.1.0) by the total number of loaded cells.

### 4.7. Dropout Probability Estimation

For each sample and separately for each dataset generated by 10x or Parse, we estimated the dropout probabilities on the down-sampled data (20,000 reads/cell). We randomly sampled 50 cells for each sample to avoid bias from the different numbers of total cells in each sample. The estimation was performed with the *scde* (1.2.1) R package [39], which models the gene expression matrix (UMI counts) as a mixture of two probabilistic processes: one is a negative binomial process which accounts for amplified mRNA, and the other is the Poisson process that accounts for dropout mRNA. Dropout probabilities for each cell were calculated with the *scde.failure.probability()* function with the default parameters. Average dropout probabilities were summarized and plotted against gene expression magnitudes (UMI counts).

### 4.8. Correlation Analysis

For each sample, we randomly sampled 7000 cells from the down-sampled data (20,000 reads/cell) which were used for the gene expression correlation analysis. Only genes expressed in all samples were included in this analysis. Gene counts of commonly expressed genes were averaged across all cells in each sample. These average gene counts were used to assess the Pearson’s correlation between the Parse and 10x data for samples H1 and H2 separately. The Pearson’s correlation coefficients were calculated using the *cor()* function in base R with ‘*method = pearson*’.

### 4.9. Platform-Specific Marker Genes

Differentially expressed genes (DEGs) between the Parse and 10x data were identified using the *FindAllMarkers()* function from the *Seurat* (4.3.0) R package with the following parameters: *only.pos = TRUE*, *logfc.threshold = 0.25*, *min.pct = 0.1*, *test.use = ‘MAST’*, *latent.vars = c(‘nFeature_RNA’)*. Up-regulated DEGs with an average log2-fold change larger than 1 and adjusted *p* value smaller than 0.01 in Parse and 10x, respectively, were identified as marker genes of each platform. The length and GC content for each gene were downloaded from BioMart (http://asia.ensembl.org/info/data/biomart/index.html, accessed on 23 December 2023).

### 4.10. Ribosomal Protein-Coding Genes

Ribosomal protein-coding genes were identified using gene symbols starting with RPS/RPL. Single-cell RNA-seq data generated with 12 different methods [20] were downloaded from https://github.com/elimereu/matchSCore2/tree/master/data (accessed on 24 December 2023).

### 4.11. Silhouette Score

The down-sampled data (20,000 reads/cell) were clustered using graph-based clustering with first 10 PCs at a resolution of 0.4, 0.5, 0.6, 0.7, 0.8, 0.9, 1.0, 1.1, 1.2, 1.3, 1.4, and 1.5 with the functions *FindNeighbors()* and *FindClusters()* in the *Seurat* (4.3.0) R package. We used the *silhouette()* function from the *cluster* (2.1.3) R package to compute Average Silhouette Width (ASW) scores based on Euclidean distances calculated by the *dist()* function in the *stats* (4.2.2) R package.

### 4.12. Cell-Type Signature Score

Bulk RNA-seq data from sorted immune cells [27] were obtained by calling the *DatabaseImmuneCellExpressionData()* function from the *celldex* (1.4.0) R package [37]. Differential expression analysis was carried out for each cell type’s RNA-seq data using the *findMarkers()* function from the *scran* (1.22.1) R package [40]. Up-regulated genes with an FDR less than or equal to 0.01 were used as cell-type gene signatures. *AUCell_calcAUC()* from the *AUCell* (1.16.0) R package [41] was used to calculate the AUC of each cell-type gene signature in each cell. The median value of AUC scores across all the cells in that cell type were summarized and are indicated in the violin plots as the measurement of dataset’s capability in cell-type annotation.

## Figures and Tables

**Figure 1 ijms-25-03828-f001:**
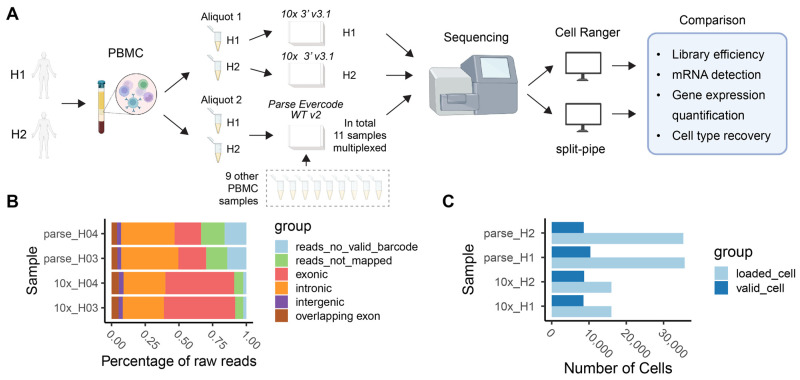
Overview of study design and library efficiency. (**A**) Two human PBMC samples, H1 and H2, were split into two aliquots. One aliquot was used for 10x (3′ v3.1) library preparation and the other for Parse (Evercode WT v2) library preparation. These were sequenced in parallel. Gene expression counts were generated separately, using pre-process pipelines specific to each method (*Cell Ranger* v7.1.0 [16] and *split-pipe* v1.0.4p). (**B**) Percentage of raw reads discarded (reads_no_valid_barcode); not mapped to the genome (reads_not_mapped); and mapped to exonic, intronic, intergenic, and overlapping-exon regions. (**C**) Number of cells loaded/input and number of valid cells output from pre-process pipelines.

**Figure 2 ijms-25-03828-f002:**
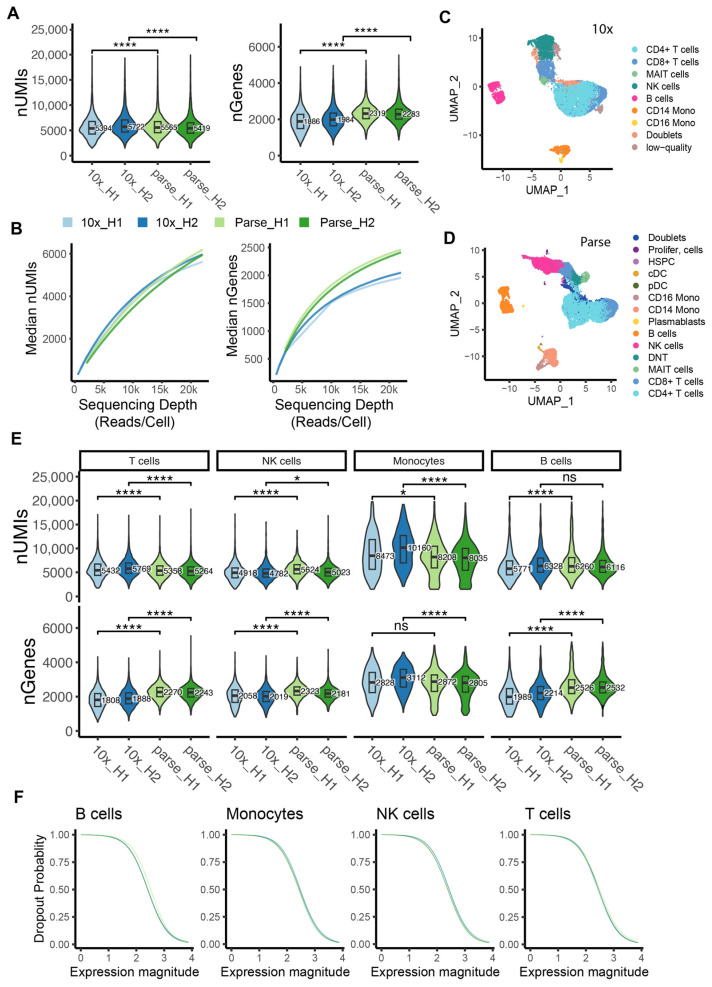
Sensitivity in capturing mRNA molecules. (**A**) Violin plot displaying the distribution of number of UMIs (left panel) and genes (right panel) per cell across all samples at a sequencing depth of 20,000 reads/cell (**** *p* ≤ 0.0001, two-samples *t*-test). (**B**) Median number of detected transcripts (UMIs) and genes (nGenes) at different down-sampled sequencing depths in each sample (**C**,**D**) UMAP visualization of cell type annotations in 10x and Parse data. (**E**) Violin plot displaying the distribution of the number of UMIs (top) genes (bottom) per cell for the major cell types across all samples at a sequencing depth of 20,000 reads/cell (ns: *p* > 0.05, * *p* ≤ 0.05, **** *p* ≤ 0.0001, two-samples *t*-test). (**F**) Dropout probabilities plotted as a function of gene expression magnitude for each method (green and blue lines, respectively) for 4 major representative cell types in PBMCs.

**Figure 3 ijms-25-03828-f003:**
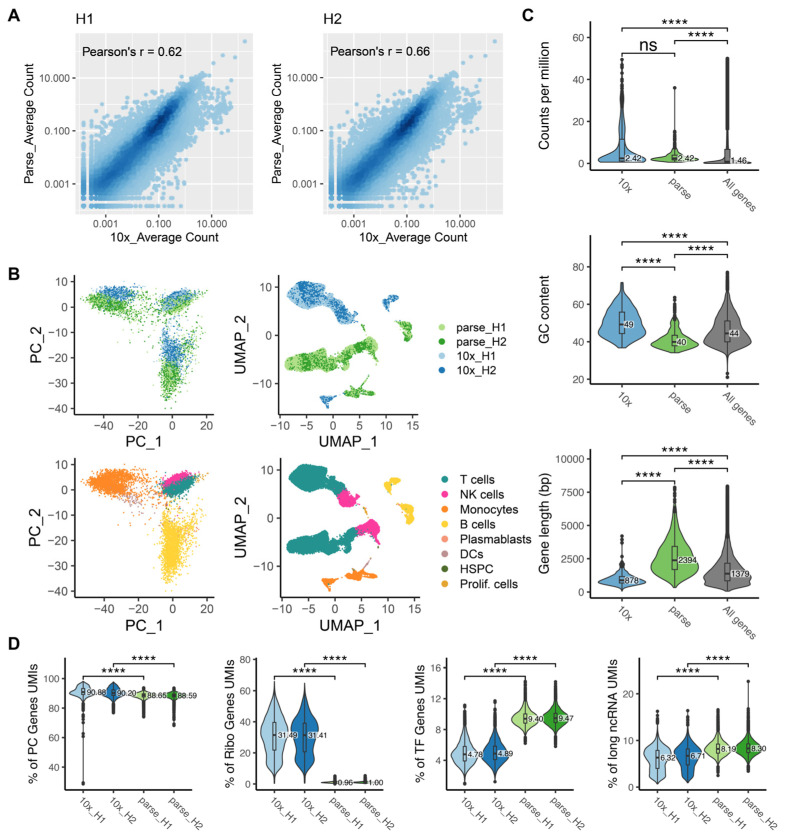
Comparative analysis of gene quantification bias. (**A**) Density scatter plots showing gene expression correlation between the two platforms in sample H1 (**left**) and sample H2 (**right**). (**B**) PCA and UMAP visualization of cells, colored by method and cell type. (**C**) Distributions of gene expression abundance, gene GC content, and gene length. Marker genes from each platform were used to demonstrate the distributions in the first two violins (317 genes for 10x; 556 genes for Parse). The distribution of all genes (29,184) is shown for comparison in the third violin. (ns: *p* > 0.05, **** *p* ≤ 0.0001) (**D**) Percentage of UMIs mapped to protein-coding (PC) genes, ribosomal protein-coding genes, transcription factor (TF) genes, and long non-coding RNAs in each cell. (ns: *p* > 0.05, **** *p* ≤ 0.0001, Wilcoxon test).

**Figure 4 ijms-25-03828-f004:**
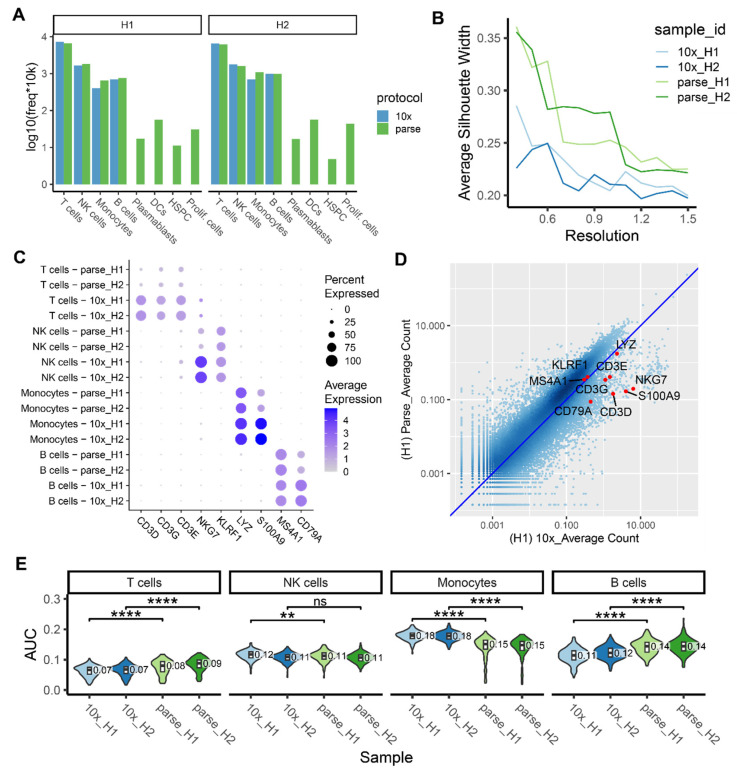
Clustering and cell type annotation. (**A**) Bar graph showing cell types identified in each method and the number of cells in each cell type. Y-axis is the number of cells detected in every 10,000 cells with log10 transformation to better visualize the proportion of rare cell types. (**B**) Clustering quality at different clustering resolutions, measured by average silhouette width (ASW) score. (**C**) Dot plot showing the expression of key cell-type markers in two methods. The size of the dot indicates the percentage of cells expressing that marker and the color indicates the average expression level. (**D**) Density scatterplot showing gene expression correlation between the two platforms in sample H1. The X-axis is the average UMI count/gene in the 10x data and Y-axis is the average UMI/gene count in the Parse data. Cell-type markers are highlighted as red dots. (**E**) Cell-type activity score calculated by *AUCell* for each cell type in each sample (ns: *p* > 0.05, ** *p* ≤ 0.01, **** *p* ≤ 0.0001, two-samples *t*-test).

**Table 1 ijms-25-03828-t001:** Summary of the comparative analysis between Parse Biosciences and 10x Genomics platforms.

PLATFORM		10x	Parse
STRENGTHS		- Higher library efficiency	- Enables tagging of 96 samples in parallel in a single batch with up to 1 M cells - Higher sensitivity in gene detection
WEAKNESSES		- Enables tagging of 16 (max) samples sequentially in a single lane with up to 10 K cells - Lower sensitivity in gene detection	- Significantly lower cell recovery rate
**ASSESSED FEATURE**	**METRICS**	**10x**	**Parse**
LIBRARY EFFICIENCY	*Cell recovery* *(average %)*	53%	27%
	*Valid reads* *(average %)*	98%	85%
SENSITIVITY IN GENE DETECTION	*Number of genes* *(median [range])*	1938 [515–4967]	2302 [860–5548]
POTENTIAL BIASES IN GENE QUANTIFICATION	*Gene length* *(median [range])*	881 bp [255–18,662]	2394 bp [331–36,798]
	*GC content* *(median [range])*	49.4% [36.8–71.6%]	39.9% [34.1–63.6%]
	*Ribosomal protein-* *coding gene abundance* *(median [range])*	30.7% [0.4–66.0%]	1.0% [0–5.3%]
ABILITY TO IDENTIFY CELL TYPES ACCURATELY	*Manual annotation* ^1^ *(average expression* ^2^ *,* *% of cells expressed* ^3^ *)*	B cells: 8.0, 93.9% Monocytes: 102.0, 99.5% NK cells: 26.8, 91.1% T cells: 3.4, 80.3%	B cells: 4.2, 78.0% Monocytes: 14.8, 74.9% NK cells: 2.5, 58.7% T cells: 0.8, 30.4%
	*Reference-based scoring* ^4^ *(median AUC [range])*	B cells: 0.12 [0.04–0.18] Monocytes: 0.18 [0.12–0.22] NK cells: 0.11 [0.04–0.16] T cells: 0.07 [0.02–0.11]	B cells: 0.14 [0.08–0.24] Monocytes: 0.15 [0.02–0.19] NK cells: 0.11 [0.04–0.16] T cells: 0.08 [0.02–0.12]

^1^ We annotated each cell cluster using the expression of known cell-type-marker genes (see Section 4). ^2^ For each cell type, we report the average expression of cell-type markers across all cells in the cluster. ^3^ Percentage of cells in the cell-type cluster that express the cell-type markers. ^4^ For each cell type cluster, we calculated a cell-type activity score (AUC) using the gene signatures from reference dataset of sorted immune cells [27] (see Section 4).

## Data Availability

The datasets generated during the current study are available in the GEO repository with GEO accession number GSE246624 (https://www.ncbi.nlm.nih.gov/geo/query/acc.cgi?acc=GSE246624, accessed on 20 January 2024).

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
