# Peer review of "Comparative Analysis of Single-Cell RNA Sequencing Methods with and without Sample Multiplexing"

_ijms, 2024, doi:10.3390/ijms25073828_

Round 1

Reviewer 1 Report

Comments and Suggestions for Authors

Review for Comparative analysis of single-cell RNA sequencing methods with and without sample multiplexing.

Due to the needs of large single-cell data from multiple samples, batch effects caused by different treatment conditions and other technical elements should be eliminated to prevent false discoveries in single-cell studies. To access performance of conventional single-cell platform and multi-sample platform, this manuscript compared the transcriptomic profiles demultiplexed from Parse data with those obtained from a single 10x library. Specifically, library efficiency, sensitivity in gene detection, potential bias in gene quantification and ability to identify cell types accurately are accessed and displayed by clear and well-ordered graphs and table. This research will assist researchers in selecting appropriate sequencing platforms for sequencing based on different sample types, thus making it meaningful. This manuscript can be considered to be published after the minor issues are fixed.

Minor issues:
1. In figure 1A, whether to use “multiplexing” should be indicated on the graph to illustrate the difference in sample processing between the two sequencing platforms.

2. In figure 2A and 2E, since the number of gene detections and UMI detections are relatively close for each method, it is best to label their medians on the graph to allow readers to clearly compare the differences.

3. Since Parse used random hexamers, in principle it should detect more non-coding RNAs. The authors should evaluate long ncRNA and short ncRNA detection capabilities between 10X and Parse.

4. Given the gene detection bias between two platforms, the authors should look into the genes showed biased detection to see whether there is a pattern besides gene length and GC content.

5. In line 364-365, the authors suggested that its Parses's tendency to detect longer and low GC content genes caused relatively lower expression level of cell type marker genes in Parse data. Yet the authors didn't provide the gene length or GC content information for those cell marker genes. The authors should either provide this information or provide another explanation.

6. This manuscript analyzed the advantages and disadvantages of different sequencing platforms and provides possible reasons for these characteristics. In the discussion section, can the authors provide a prospect for a possible sequencing method that combines the advantages of these two platforms? I believe this will make the discussion section of the article more comprehensive.

Reviewer 2 Report

Comments and Suggestions for Authors

Methodology Clarification: Elaborate more on the sample preparation and the exact steps followed in each sequencing method. This would help better understand the differences between the methods and the replicability of the experiments.

Deeper Statistical Analysis: Although the study compares library efficiency, gene detection sensitivity, and the ability to recover distinct cell clusters, a more robust statistical analysis could strengthen the results. For instance, specific statistical tests demonstrating significance between observed differences could provide a more solid basis for the conclusions.

Discussion on Limitations: A more in-depth discussion of the limitations of each platform could be beneficial. For example, addressing limitations in detecting rare cell types or quantifying low-abundance gene expression could offer a more complete view of the performance of each method.

Comments on the Quality of English Language

The English quality in the manuscript, based on the sections reviewed, appears to be of a high standard, typical of academic writing in the field of molecular biology and genomics. The text demonstrates a clear understanding of complex scientific concepts and presents these using appropriate technical terminology. Sentences are well-constructed, and the use of language is precise, which is essential for conveying detailed scientific methods and findings accurately.

Reviewer 3 Report

Comments and Suggestions for Authors

In this research article, the authors are investigating the effect of multiplexing in single cell sequencing. They used two conceptually different methods and compared their performance in PBMCs in terms of cell recovery rate, library preparation performance, number of genes detected and finally cell clustering and rare cell type detection. Overall, the article is interesting and the topic is currently of high interest. However, the article could benefit from some more precision:

11)      I could not find any reference to the duplicate rate in this article, could the authors comment on whether amplification in both systems resulted in a similar duplicate rate? Were duplicates removed prior to the presentation of the read grouping in Figure 1B?

22)      Can the authors provide a list of genes detected only by Parse and not 10x (and vice versa) in a supplement or in the Geo repository? Are these genes enriched for particular types of RNA? Can they be related to the random hexamer?

33)      Could the authors add to the analysis of the ribosomal content in the supplementary analysis the analysis of GC content in the different single cell methods? It would be interesting to know how different methods compare with 10x and Parse.

44)      Figure 4 it is not easy to read. Am I correct in calculating that the authors detected 10 Plasmblasts, 10 HSPC, 50 DC and 50 proliferative cells per 10.000 cells? If so, is this number of cells relevant in any particular analysis?

55)      PBMC were frozen, then thawed, FACSed and prepared for either 10x or Parse. Given that the cells were fixed for Parse, and given that 10x metrics may reflect higher stress on the cells during processing, do the authors think that the fixation may have a positive effect on the performance of Parse?
